# Mobile phones carry the personal microbiome of their owners

James F. Meadow[1], Adam E. Altrichter[1] and Jessica L. Green[1,2]

[1] Biology and the Built Environment Center, Institute of Ecology and Evolution, University of Oregon, Eugene, OR, USA
[2] Santa Fe Institute, Santa Fe, NM, USA

## ABSTRACT

Most people on the planet own mobile phones, and these devices are increasingly being utilized to gather data relevant to our personal health, behavior, and environment. During an educational workshop, we investigated the utility of mobile phones to gather data about the personal microbiome — the collection of microorganisms associated with the personal effects of an individual. We characterized microbial communities on smartphone touchscreens to determine whether there was significant overlap with the skin microbiome sampled directly from their owners. We found that about 22% of the bacterial taxa on participants' fingers were also present on their own phones, as compared to 17% they shared on average with other people's phones. When considered as a group, bacterial communities on men's phones were significantly different from those on their fingers, while women's were not. Yet when considered on an individual level, men and women both shared significantly more of their bacterial communities with their own phones than with anyone else's. In fact, 82% of the OTUs were shared between a person's index and phone when considering the dominant taxa (OTUs with more than 0.1% of the sequences in an individual's dataset). Our results suggest that mobile phones hold untapped potential as personal microbiome sensors.

## INTRODUCTION

Few human possessions are so universally owned as mobile phones. There are almost as many mobile phones as there are humans on the planet (*Cisco Systems Inc., 2014*). More people worldwide own mobile phones than have access to working toilets (*WHO/UNICEF Joint Monitoring Programme for Water Supply and Sanitation, 2013*). These devices not only help individuals share information with each other, they are increasingly being used to help individuals gather information about themselves. Smartphones — mobile phones with built in applications and internet access — have rapidly become one of the world's most sophisticated self-tracking tools. Self-trackers and those engaged in the "quantified self" movement (*Swan, 2013*) are using smartphones to collect large volumes of data about their health, their environment, and the interaction between the two. Continuous tracking is now obtainable for personal health indicators including physical activity (*Hirsch et al., 2014*), brain activity (*Stopczynski et al., 2014*), mood dynamics (*Morris et al., 2010*),

Corresponding author
James F. Meadow,
jfmeadow@gmail.com

numerous physiologic metrics (*Boulos et al., 2014*) and demographic data (*Palmer et al., 2013*). Similarly, smartphones are empowering individuals to measure and map, at a relatively low cost, environmental data on air quality, water quality, temperature, humidity, noise levels, and more (*McGrath & Scanaill, 2013*).

Mobile phones can provide another source of information to their owners: sample data on their personal microbiome. The personal microbiome, here defined as the collection of microbes associated with an individual's personal effects (i.e., possessions regularly worn or carried on one's person), likely varies uniquely from person to person. Research has shown there can be significant interpersonal variation in human microbiota, including for those microbes found on the skin (*Fierer et al., 2008*; *The Human Microbiome Project Consortium, 2012*; *Grice et al., 2009*; *Grice & Segre, 2011*). We hypothesize that this variation can be detected not just in the human microbiome, but also on the phone microbiome. The mobile phone is a personal effect so regularly carried it has been described in popular culture as 'an extension of self'. The reality that many people take their phones with them everywhere they go suggests that, at any given point in time, phones and their owners are exposed to similar environmental microbiota, which can lead to overlapping microbiota composition. The frequency with which people directly touch their phones provides an additional mechanism leading to shared microbiota composition. Research has shown that traces of human microbiota are left in rooms we occupy and on surfaces we touch (*Flores et al., 2011*; *Hospodsky et al., 2012*; *Flores et al., 2013*; *Meadow et al., 2013*; *Dunn et al., 2013*; *Meadow et al., 2014*). In some cases, these microbial signatures can be attributed to individual people (*Fierer et al., 2010*). Thus mobile phones presumably carry a strong signal of their owner's human microbiome, and might be identifiable to that owner.

To explore these possibilities, we collaborated with participants in an educational workshop centered on technologies and tools commonly used in microbiome research. We started with the basic hypothesis that bacterial communities on mobile phones will reflect the microbiome of a frequent human contact point: fingers. Previous studies have reported that some of the bacteria found on mobile phones can also be found on the owner's hands, however these studies have been limited to culture-based methods in a medical setting and thus reflect patterns for a relatively small fraction of microbial life (e.g., *Brady et al., 2009*; *Walia et al., 2014*). If a hand-phone microbiome connection holds for entire bacterial communities, mobile phones can potentially be a non-invasive way to track environmental microbial exposure over time and space, and inform how we exchange human microbiota with our immediate surroundings.

As a first step towards understanding the hand-phone microbiome connection, we asked whether hand-associated microbiota can be detected on a phone, and whether this detected connection is reliably strongest for the owner of that mobile phone (as opposed to people who do not regularly touch the phone). We explored this question with workshop participants, who sampled the microbiota on their index fingers and thumbs, as well as the touchscreens of their own smartphones. We analyzed these bacterial assemblages to answer three questions: (1) Are the bacteria on mobile phones indicative of frequent human contact? (2) Do women and men differ in their microbial connections to their phones?

(3) Are the bacterial communities sampled from phones more similar to their owners than to other people?

## MATERIALS AND METHODS

### Ethics statement

All participants were fully informed of the nature of the educational workshop. Research protocols were reviewed and ruled exempt under the University of Oregon Institutional Review Board (45 CFR 46.101(b)(4)). Participants' identities are unknown and were never recorded during sampling.

### Sample collection

All samples were collected during a workshop at the Robert Wood Johnson Foundation in Princeton, New Jersey, May 21, 2013. Surfaces were sampled using a Copan nylon flocked swab (www.copanusa.com, Murrietta, CA, USA; item #551C) moistened with sterile buffer solution (0.15M NaCl, 0.1M Tween20). Seventeen volunteer participants sampled the touch-surfaces of their own mobile phone, as well as their own thumb and index finger on their dominant hand (3 samples for each of 17 participants). Each surface was swabbed for approximately 20 s. Each subject was asked to mark their tubes with the following information: gender, dominant hand, and whether or not they had washed their hands in the last hour. Swabs were placed back in the original sterile tubing, sealed, frozen, and shipped back to the University of Oregon. All samples were stored at −80 °C until processed.

### 16S sequencing library prep

DNA was extracted from 3 swabs per person for 17 different individuals. Samples were extracted using the Extract-N-Amp Plant PCR kit (Sigma Aldrich). Each swab tip was added to 100 µl of extraction solution, heated for 10 min at 95 °C, and 100 µl of dilution solution added. Finally, the V4 region of bacterial 16S rRNA gene was amplified and prepped for amplicon sequencing (*Flores, Henley & Fierer, 2012*).

Sequencing libraries were prepped using a modification of *Caporaso et al. (2012)* protocol where 16S rRNA gene primers 515F and Golay-barcoded 806R were used in triplicate PCRs per sample, followed by equivolume combination of all samples, and concentrated to 25 µl (Zymo Research Clean and Concentrate-5). This was followed by gel electrophoresis size selection and extraction of the pooled samples (Qiagen MinElute Gel Extraction), and a final clean up step (Zymo Research Clean and Concentrate-5). The PCR had the following components (20 µl total volume): 5 µl PCR-grade water, 10 µl Extract-N-Amp PCR ReadyMix (contains polymerase and dNTPs), 0.5 µl each primer (10 µM), and 4 µl of genomic DNA template. The PCR was carried out under the following conditions: an initial denaturation step of 94 °C for 3 min, followed by 35 cycles of 94 °C for 45 s, 52 °C for 45 s and 72 °C for 35 s, with a final extension at 72 °C for 10 min. The final library was then sent to the Dana-Farber Cancer Institute Molecular Biology Core Facilities for 250 PE sequencing on the Illumina MiSeq platform.

## Data processing and statistical analysis

Raw sequences were processed using the QIIME v. 1.7 pipeline (*Caporaso et al., 2010*). We retained and demultiplexed $3.2 \times 10^6$ forward-read sequences with an average quality score of 30 over 97% of the sequence length. Sequences were binned into operational taxonomic units (OTUs) at 97% sequence similarity using UCLUST *de novo* clustering (*Edgar, 2010*), which resulted in 34,400 OTUs across 56 samples. Taxonomy was assigned to OTUs using the RDP classifier and Greengenes version '`4feb2011`' core set (*DeSantis et al., 2006*).

After quality filtering, demultiplexing, and OTU clustering, all statistical analyses were conducted in `R` (*R Development Core Team, 2010*), primarily with the `vegan` and `labdsv` ecological analysis packages (*Oksanen et al., 2011*; *Roberts, 2012*). Plant chloroplast sequences and mitochondrial sequences were removed by name (e.g., taxonomic classifications were queried for phylum = '`Streptophyta`' or genus = '`mitochondria`'). OTUs represented by fewer than 3 sequences were also removed to eliminate potential sequencing anomalies. We also removed 3 OTUs that were highly abundant in negative sequencing controls (blank-template samples from the Extract-N-Amp PCR kit); that decision and its analytical implications are detailed in Data S1. All samples were rarefied to 7,000 sequences per sample to achieve approximately equal sampling depth.

We used two different approaches to measure the shared bacterial communities among phones and fingers. We calculated OTU turnover using the Jaccard taxonomic metric, and conducted discriminant analysis using the Canberra taxonomic metric. In other words, barplots displaying shared percentages of OTUs were constructed with Jaccard similarities since values resulting from this metric are easily translatable as the percent of shared OTUs between any two samples. Ordinations and multivariate tests, on the other hand, used Canberra distances, since this metric incorporates abundance data and emphasizes contributions from relatively rare OTUs. Ordinations were constructed using iterative non-metric multi-dimensional scaling (NMDS). Community differences were assessed using permutational multivariate analysis of variance tests (PERMANOVA). Since PERMANOVA community differences were tested with 999 permutations, we report resulting *p*-values down to, but not below, 0.001. Jaccard similarities (visualized in the gray bar plot figures) were compared with *t*-tests, and those distances were paired every time individuals were compared to others. A reproducible record of all statistical analyses is included as Data S2, and is available on GitHub (https://github.com/jfmeadow/Meadow_etal_Phones). This includes all underlying data and R code for all analyses, figures and tables. This dynamic analysis record was created using the R package `knitr` (*Xie, 2013*). Raw data have been deposited in the open-access data repository figshare (*Meadow, 2014*).

## RESULTS

After sequence processing and rarefaction to even sampling depth, we analyzed c. $3.57 \times 10^5$ sequences representing 7,404 bacterial OTUs from 51 samples. The most consistently abundant OTU over the whole dataset (median = 26.36% of sequences) was 100% similar to several *Streptococcus* spp. that are common human oral residents (e.g., *S. oralis*, NCBI accession number NR102809; *S. mitis*, NR102808; and *S. infantis*,

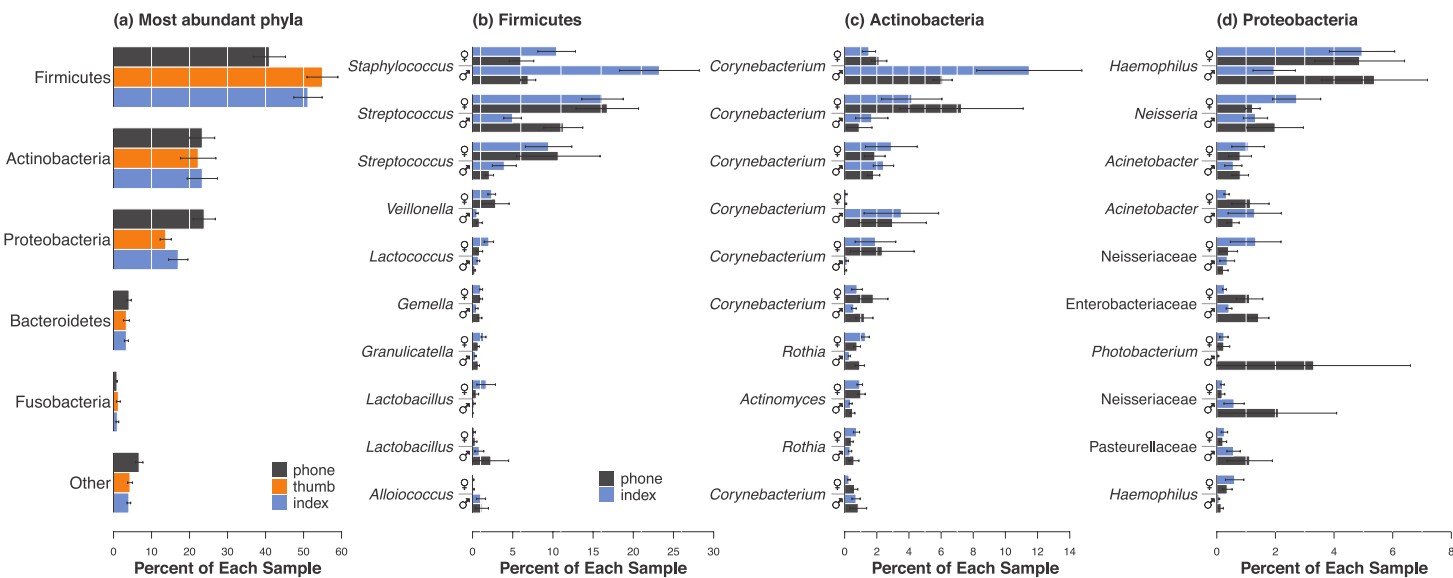

**Figure 1** **Differences in the most abundant phyla were apparent among phones, fingers, and between men and women.** (A) The five most abundant phyla represented in this study were generally consistent regardless of surface type. Within the top three phyla, Firmicutes (B), Actinobacteria (C), and Proteobacteria (D), several prominent OTUs were differentially abundant either on phones or by gender. Bacterial groups are ordered top to bottom by overall mean relative abundance. Values are shown as mean relative abundances ±1 standard error. OTU names in (B), (C) and (D) are GreenGenes-assigned genera unless they were only defined to family resolution. Repeated names indicate that multiple OTUs within the same genus were among these most abundant. Since thumbs and index fingers yielded similar results, only phones and index fingers (black and blue, respectively) are shown in (B), (C) and (D).

NR042928). The second most abundant OTU (median = 23.04% of sequences) was 100% similar to several *Staphylococcus* spp. that are well known as human skin-associated microbes (e.g., *S. warneri*, NR102499; *S. aureus*, NR075000; and *S. epidermidis*, NR074995). These two OTUs were also the two most abundant for all groups of samples we considered (i.e., phones, fingers, women and men), except that the second most common OTU detected in men's samples was 100% similar to *Corynebacterium tuberculostearicum* (NR028975). We also found that men, women, phones, and fingers all showed a few strong differences in *mean* abundance across each specific group (Fig. 1). Given that the two fingers yielded qualitatively similar results, we report most results just for index fingers hereafter to represent either finger that frequently contacts a phone.

## Are bacterial communities on mobile phones indicative of frequent contact with their owners?

Consistent with previous culture-based studies that have reported pathogen transfer between hands and phones (*Brady et al., 2009*; *Walia et al., 2014*), we found that several of the most abundant bacterial OTUs present on phones are also commonly associated with human body habitats. For this study, we assumed that thumbs and index fingers are the two digits that people use most when interacting with their phone, so we sampled both fingers on the dominant hand from all participants to determine which is the best proxy for mobile phone connectedness. The two fingers from each participant shared, on average,

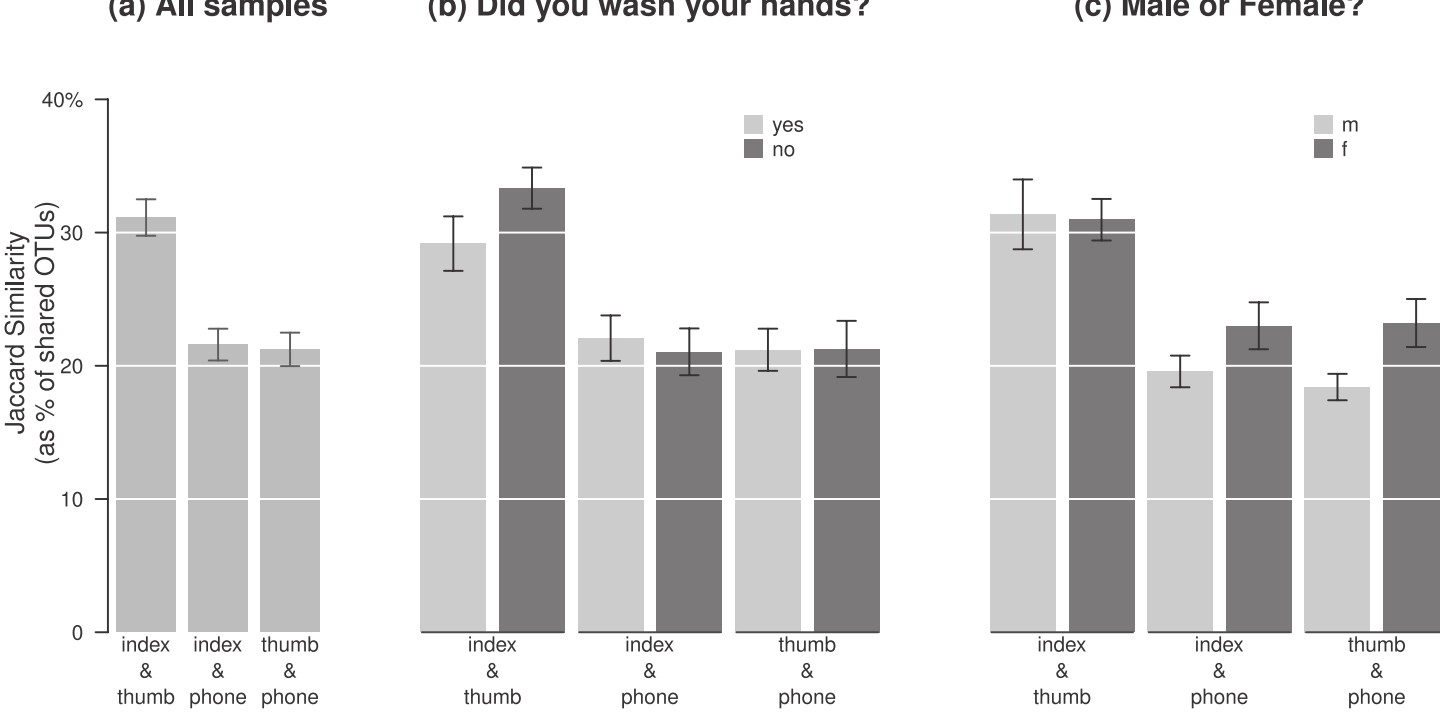

**Figure 2** **The degree of overlap between bacterial pools differed by gender and by whether participants washed their hands.** (A) Thumbs and index fingers on the same person shared, on average, 32% of their OTUs, while each digit shared about 22% with their own phone. The two fingers had significantly more in common than either did with phones ($p < 0.001$ for both fingers; from paired $t$-tests comparing the 1st bar in (A) to the 2nd and 3rd, respectively). (B) Hand-washing made a marginal but insignificant difference in the resemblance of the two fingers ($p = 0.126$; comparing the 1st and 2nd bars in (B)), and no difference at all in the finger/phone connection ($p = 0.7$; comparing the 3rd and 4th bars in (B)); (C) Women's fingers appeared to share more OTUs with their phones than men, but the difference was not significant ($p = 0.128$; comparing the 3rd and 4th bars in (C)) since both shared more OTUs, on average, with their own phones than with anyone else's (Table 1).

32% of OTUs, while both fingers shared about 22% of OTUs with their respective phones (Fig. 2A).

Participants who had not washed their hands within the previous 60 min (Fig. 2B) shared 4% more bacteria among their two digits than those who had, though the difference was not significant ($p = 0.126$), nor was the difference in the pool of bacterial OTUs shared with their mobile phones ($p = 0.7$).

When we limited OTUs to only those representing more than 0.1% of a single person's dataset, we found that, on average, 82% of OTUs were in common between index fingers and phones, while 96% of OTUs were shared between index fingers and thumbs.

## Do women and men differ in their microbial connections to their mobile phones?

Men and women exhibited significant differences in bacterial communities, regardless of whether considering phones, either or both fingers, or all samples together (Table 1 and Fig. 3). This difference is evident in several of the top *Corynebacterium* OTUs (Fig. 1C). Although men and women both shared bacteria with their own phones, women appeared to have a stronger microbiological connection to their phones than men

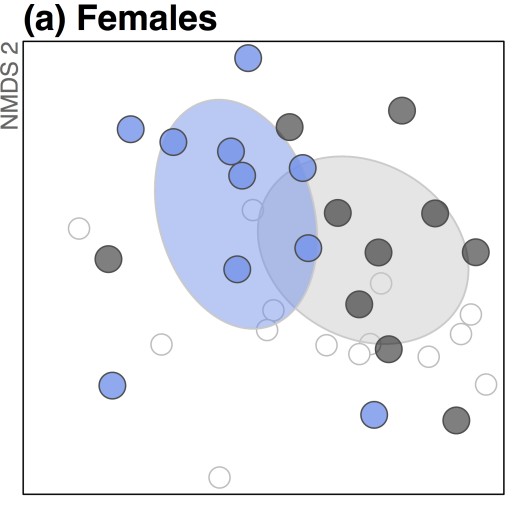
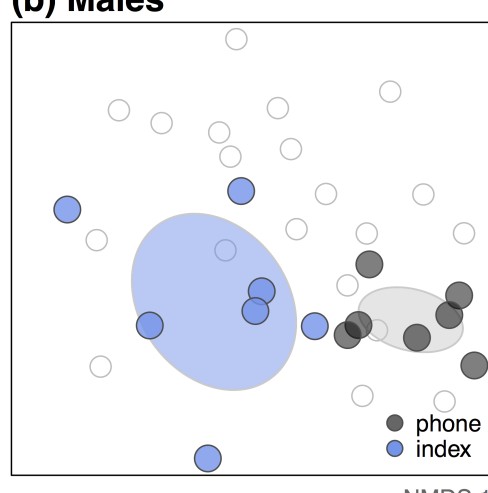

**Figure 3** **As a group, women (A) more closely resembled their phones than did men (B).** Gender differentiated both mobile phones ($p = 0.007$) and index fingers ($p = 0.033$). Considered as a group, women's index fingers were not significantly different from their phones ($p = 0.327$), but men's were ($p = 0.001$). In other words, the 95% confidence intervals for each group of index finger bacterial communities (blue ellipses) and mobile phone bacterial communities (gray ellipses) overlap significantly for women, but not for men. All points displayed in (A) and (B) are part of the same ordination, but highlighted by gender.

(Figs. 2C and 3). For women, the bacterial community composition on their index fingers was not significantly different from that sampled on their mobile phones ($p = 0.327$; Fig. 3A). For men, there was a significant difference ($p = 0.001$; Table 1 and Fig. 3B).

**Are bacterial communities on phones identifiable to their owners?**

To further explore the personal microbiome connection, we asked a basic question: does your phone resemble you? Or put another way, does your own phone carry microbes that resemble your own microbiome more so than another person's microbiome? To answer this question, we compared the number of OTUs each person shared with his/her own phone to all other phones in the study. We found that yes, an individual's finger shared on average 5% more OTUs with his or her own phone than with everyone else's phones ($p < 0.001$; Fig. 4 and Table 1). We found this to be the case for both men and women, even though communities on men's phones were overall significantly different from their hands (as in Fig. 3).

## DISCUSSION

In this study, we explored the potential for mobile phones to monitor the personal microbiome — the collection of microorganisms inhabiting an individual's personal effects. Considering that mobile phone users regularly carry their phones and touch them on average 150 times per day (*Meeker & Wu, 2013*), one might expect to find substantial overlap between the microbiota sampled from phones and their owners' fingers. We found

**Table 1 Men and women differed in their bacterial communities, and women more closely resembled their phones than men.** All tests except the last 3 were from permutational multivariate analysis of variance (PERMANOVA) with 999 permutations. The final tests (bold) were from a paired *t*-test comparing each person's similarity with his/her own phone to his/her average similarity to all other phones. DF, residual degrees of freedom for each statistical test.

| Test | DF | *P*-value | Reference Figure |
|---|---|---|---|
| Do men and women differ in their bacterial communities? | | | |
|     all samples | 49 | 0.001 | – |
|     only phones | 15 | 0.007 | 3 |
|     both fingers | 32 | 0.002 | – |
|       index fingers | 15 | 0.033 | 3 |
|       thumbs | 15 | 0.027 | – |
| Are women's index fingers (as a group) different from their phones? | 18 | 0.327 | 3A |
| Are men's index fingers (as a group) different from their phones? | 12 | 0.001 | 3B |
| **Do you share more with your phone than with other phones?** | | | |
|     **all participants** | **16** | **<0.001** | **4** |
|     **only women** | **9** | **0.005** | **4** |
|     **only men** | **6** | **0.049** | **4** |

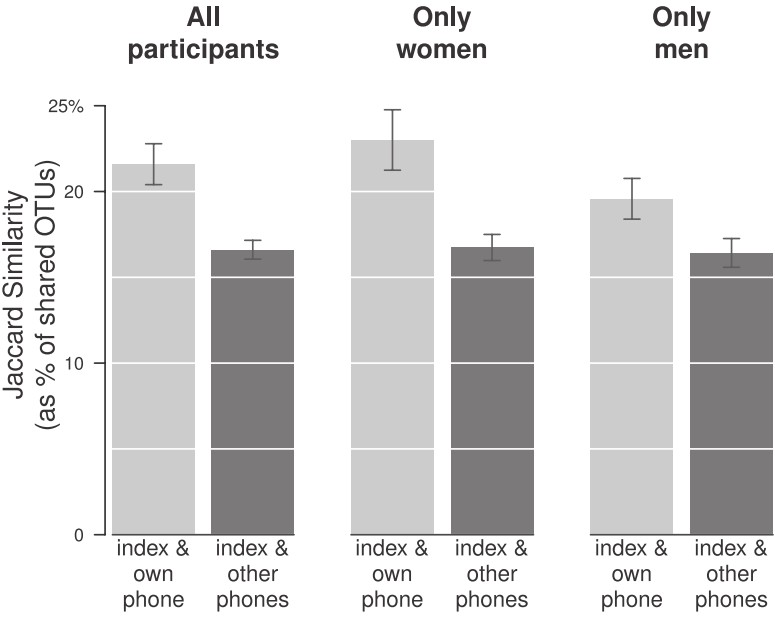

**Figure 4 People shared more bacterial OTUs with their own phones than with phones belonging to other people.** Each person's index finger was compared to his/her own phone and then to their average Jaccard similarity with other people's phones (leftmost bars; $n = 17$; $p < 0.001$). The same was true for both sexes, though the effect was stronger for women (middle two bars; $n = 10$; $p = 0.005$) than for men (rightmost bars; $n = 7$; $p = 0.49$).

this expectation to hold: on average, 22% of the bacterial OTUs on participants' fingers were also on their own phones. And although when considered as a group, women and men differed in the extent of their biological connections to their phones, when considered on an individual basis, both men and women shared more bacterial taxa with their own phones than with anyone else's phone.

The pool of ≈22% of OTUs that overlap between our phones and fingers (compared to 32% shared between two fingers on the same hand) might seem surprisingly low since many people tend to regularly carry and touch their phones. However, 82% of the most common OTUs (those representing more than 0.1% of a single person's dataset) were shared between phones and fingers, while 96% were shared between index fingers and thumbs. As mentioned above, our fingers are only one plausible source for microbes on our phones. They may also be dispersed in from other parts of our body (e.g., the palm, mouth, face, or ears), our clothes and belongings, the people we interact with, and our surrounding built environment. Thus phones might be valuable for chacterizing exposure to biological threats or unusual sources of environmental microbes that don't necessarily end up integrated into our human microbiome.

Dispersal from external sources might not be the only ecological process determining which microbial assemblages we detected on mobile phones. It is possible that the surface of a phone acts as an environmental filter that selects for only a subset of the bacterial taxa it is frequently exposed to. Environmental conditions including material type, temperature, pH, moisture, ultraviolet light exposure, and substrate availability are a few examples of what might influence the structure and dynamics of the communities on phones. That being said, the high degree of overlap in the most abundant OTUs on fingers and phones suggests that dispersal is plausible as a major driving factor.

Our findings are largely consistent with previous research. Several studies have reported that some of the same bacteria found on mobile phones are also found on their owner's hands or other body parts (*Brady et al., 2009*; *Brady et al., 2011*; *Ulger et al., 2009*; *Pal et al., 2013*; *Beckstrom et al., 2013*; *Kiedrowski et al., 2013*). All of these studies, however, used culture-based methods to target pathogenic bacteria, primarily in healthcare settings. While cell phones have been suggested as "Trojan horses" for pathogenic infection (*Walia et al., 2014*), there is no direct evidence that pathogens on mobile phones influence the rate of hospital acquired infections (*Tacconelli, 2011*; *Manning et al., 2013*), or that mobile phones present any more infection risk than any other human possession. Since we employed short-read 16S sequencing, and this method is unsuitable for strain-specific pathogen detection, we did not assess mobile phone pathogen risk potential in the present study. We sampled from ostensibly healthy participants at an educational workshop in a non-healthcare setting, so there was no reason to assume pathogen risks during our sampling. Our results indicate that there are thousands of bacterial taxa present on mobile phones, and suggest that our phones, and likely other possessions, carry a detectible extension of our own human microbiome — our personal microbiome.

Our study has several limitations worth considering. The sample size was modest, and the study was designed and conducted as a teaching exercise. Additionally, we only

considered mobile phones that had touchscreens (smartphones) as opposed to those with keypads, even though this difference has been shown to influence the bacteria recovered in culture-based experiments (*Pal et al., 2013*). Nor did we distinguish hand-washing methods that also might influence results (*Goldblatt et al., 2007*). That being said, our study demonstrates the value of further research into using mobile phones as personal microbiome sensors.

The link between the human microbiome and health is complex and still poorly understood (*Finucane et al., 2014*). As our understanding of this link increases, however, microbiome monitoring, diagnosis and treatment will potentially become common in medical practice. In this context, noninvasive sampling of personal effects, like mobile phones, might be a useful tracking strategy. The implications for healthcare workers are more obvious. Early detection of nosocomial infection risk through biological phone monitoring could potentially improve prevention. It remains to be seen whether daily interactions, for instance with infected patients, can be detected on mobile phones, and whether that detection can be reliably used for infection control.

Although direct human sampling is a more reliable way to collect information about the human microbiome, we can envision three future scenarios where data on the personal microbiome might be utilized: (1) possessions such as phones are easily and non-invasively sampled in large-scale microbial studies where regulatory limitations inhibit direct human sampling; (2) real-time sequencing technology can be harnessed to screen hospital visitors or health-care workers for pathogens to avoid repeated invasive human sampling; and (3) sequencing technology far better than what we have employed here will be relatively compact and cheap, such that 'real time' personal microbiome monitors can help us understand and regulate exchange between our own constituent microbes and those in our immediate environment. Further work is clearly needed to utilize personal microbiome data, but our results indicate that such innovative research directions will advance knowledge about the interactions between our human microbiota with the world around us.

Our human microbiome travels with us everywhere we go. We constantly transfer microbes to and from the surfaces around us, and that includes our possessions. We also increasingly carry our phones with us everywhere we go, and this study confirms that we share more than an emotional connection with our phones — they carry our personal microbiome.

## ACKNOWLEDGEMENTS

We thank participants of the Robert Wood Johnson Foundation's "What's Next Health?" educational workshop for their participation and input in this study. We also thank two peer reviewers for their helpful comments.

### Funding

This project was funded by a grant from the Alfred P Sloan Foundation to the Biology and the Built Environment (BioBE) Center (#2013-6-04). JLG was funded by a John Simon

Memorial Guggenheim Foundation Fellowship and by a Blaise Pascale International Research Chair funded by State and the Ile-de-France and managed by the Foundation of the Ecole Normale Supérieure. The funders had no role in study design, data collection and analysis, decision to publish, or preparation of the manuscript.

**Grant Disclosures**
The following grant information was disclosed by the authors:
Alfred P Sloan Foundation: #2013-6-04.

**Competing Interests**
Jessica L. Green is an Academic Editor for PeerJ and the Founder of Bioinformed Design, LLC.

**Author Contributions**
- James F. Meadow conceived and designed the experiments, analyzed the data, wrote the paper, prepared figures and/or tables, reviewed drafts of the paper.
- Adam E. Altrichter conceived and designed the experiments, performed the experiments, analyzed the data, wrote the paper, reviewed drafts of the paper.
- Jessica L. Green conceived and designed the experiments, performed the experiments, contributed reagents/materials/analysis tools, wrote the paper, reviewed drafts of the paper.

**Human Ethics**
The following information was supplied relating to ethical approvals (i.e., approving body and any reference numbers):

All participants were fully informed of the nature of the educational workshop. Research protocols were reviewed and ruled exempt under the University of Oregon Institutional Review Board (45 CFR 46.101(b)(4)). Participant identities are unknown and were never recorded during sampling.

**DNA Deposition**
The following information was supplied regarding the deposition of DNA sequences:

All sequencing data and accompanying sample metadata have been deposited in Figshare: http://dx.doi.org/10.6084/m9.figshare.1000786).

**Data Deposition**
The following information was supplied regarding the deposition of related data:

All scripts and analysis are available through GitHub: https://github.com/jfmeadow/Meadow_etal_Phones.

**Supplemental Information**
Supplemental information for this article can be found online at http://dx.doi.org/10.7717/peerj.447.

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
