# Peer review of "Mobile phones carry the personal microbiome of their owners"

_PeerJ, doi:10.7717/peerj.447_

## Round 0.1 · original submission · Minor Revisions

This is a nice inovative approach on the human microbiome. Please attend all the reviewers suggestions.

·

Basic reporting

The article is clear and well-written. A few minor questions and comments below:

Line 105 has bolding of the "share" in "figshare" which should be removed.

In Figure 1, the legend is missing for parts (b) and (c) (what is blue and what is black?)

Likewise in Figure 2, the color legend is missing in part (c).

In Table 1, please define "DF" in the caption of the table (just in case).

Line 146 reads "and individual's finger" should be "an individual's finger"

Line 147 should read "everyone else's phones" to remove the ambiguity with fingers.

On line 161 it seems worth mentioning the face/ears in addition to palm or mouth since phones spend a lot of time in contact with the face/ears.

Experimental design

Line 66 reads "equivolume" combination of all samples. Is this really true? Normally it's an equimolar combination to normalize for different levels of input DNA?

In line 88 the authors discuss removing "all OTUs that also showed up in negative sequencing controls". More detail would be helpful here. Firstly, what sort of controls? Water or actual DNA extraction kit controls? Next, how many OTUs got removed? How many reads were present in the negative controls? Lastly, it seems that removing all those OTUs from analysis could be overly conservative. It seems reasonable to assume that contamination in say PCR reagents and DNA extraction kits could be human-associated taxa, same as what we might find on cell phones. If only a few taxa, representing a few hundred reads were removed then it seems reasonable to remove them. But if those OTUs represent a significant fraction of the sequences from the samples of interest then this seems problematic. More information would help the reader decide whether removing all these OTUs is appropriate.

Validity of the findings

The authors have conducted an interesting, well-designed study and have produced robust data, supported by detailed statistical analysis.

However, there's a theme throughout the paper that I would like to see some clarification on. This is the idea as stated in the abstract that "mobile phones hold untapped potential as personal microbiome sensors". It's unclear to me why there is any reason not to just swab people directly? After all, the authors demonstrate that cell phones only share ~20% of their taxa with the owners fingers. This would be much better than nothing, but much worse than just swabbing the owner. The authors use the term "noninvasive sampling" to address this point... but it seems that swabbing someone's fingertip is no more invasive than swabbing their phone? Another possibility is that there is a regulatory difference between swabbing a person and swabbing their phone. The fact that the entire study came under an IRB waiver implies that there isn't a difference? It can't be both true that swabbing phones is better than swabbing people (in a regulatory sense) and then there's not an important difference (in a biological sense). The only other thing I can think of is that people feel differently about swabbing themselves versus their phones and that this would provide a way to overcome reluctance about monitoring, in say a hospital setting.

Reviewer 2 ·

Basic reporting

The paper entitled 'Mobile phones reflect the personal microbiome of their owners' by Meadow et al. is a clear, concise and well-written piece. The main result though seems pretty obvious.

Experimental design

The methods are sound, and the statistics and visualization are appropriate.

Validity of the findings

No Comments.

Additional comments

A few general comments:
- Title and throughout the text: Instead of "microbiome" it would be helpful that the authors specify "skin microbiome" as this is the habitat they are exploring.

- Figure 3: The closer resemblance between finger and phone for women might be tentatively explained by a higher proportion of resistent spore formers (Firmicutes and Actinobacteria) in women's skin microbiome (¿?). That information does not seem obvious from Fig. 1.

- Although the authors explain in the Discussion why they didn't do any pathogen detection (and I agree that strain specificity is unfeasible); I still think that showing some potential pathogen transmission would greatly enhance the manuscript. The manuscript is concise and very-well designed, so my opinion is that a little bit of speculation is acceptable. The interesting aspect of this kind of manuscripts is the potential medical issues (the results as they are now seem pretty obvious and incidental). Even the authors end up the Discussion section with medical implications. I would recommend something like assessing the abundance of potential (99% similarity, for instance) skin pathogens or the abundance of potential fecal bacteria on hand vs. phone or hand-washing vs. no hand-washing. The same group did something similar in Kembel et al. (2012) Architectural design influences the diversity and structure of the built environment microbiome

Specific comments:
- L14: Phones have also been used to track human mobility and demography. See Palmer et al. (2013) New Approaches to Human Mobility: Using Mobile Phones for Demographic Research

- L34: The statement that phones might be used as a non-invasive sampling device seems too far reaching. The comment of "inform how we exchange..." seems enough.

- Fig. 2C: Although from the main text it seems clear the categories (Male or Female), it would be helpful to include the legend as in Fig. 2B.

- L131: I think it is ok just to report the results from the index finger but just for reviewer's curiosity: Was Lactococcus also the most common OTU on the thumb?

- L154: "OTUs" instead of "taxa".

- L159: "tend to touch".

- L179: This last sentence of the paragraph does not seem necessary. It might seem obvious but the authors did not test the statement.

---

## Round 0.2 · accepted · Accept

This is a very interesting paper and the concept of personal microbiome is new and should be discussed by the research community, all the queries given by the reviewers where fully addressed, it should be ready for publication.